# Knockdown of the snoRNA-Jouvence Blocks the Proliferation and Leads to the Death of Human Primary Glioblastoma Cells

**DOI:** 10.3390/ncrna11040054

**Published:** 2025-07-18

**Authors:** Lola Jaque-Cabrera, Julia Buggiani, Jérôme Bignon, Patricia Daira, Nathalie Bernoud-Hubac, Jean-René Martin

**Affiliations:** 1Institut des Neurosciences Paris-Saclay (Neuro-PSI), CNRS, UMR-9197, Université Paris-Saclay, Campus CEA Saclay, 151 Route de la Rotonde, Batiment 151, 91400 Saclay, France; lolajaquecabrera@gmail.com (L.J.-C.); julia.buggiani@i2bc.paris-saclay.fr (J.B.); 2Institut de Chimie des Substances Naturelles (ICSN), CNRS, UPR-2301, Université Paris-Saclay, 91198 Gif-sur-Yvette, France; jerome.bignon@cnrs.fr; 3Laboratoire de Mécanique des Contacts et des Structures (LaMCoS), INSA Lyon, CNRS, UMR5259, 69621 Villeurbanne, France; patricia.daira@insa-lyon.fr (P.D.); nathalie.bernoud-hubac@insa-lyon.fr (N.B.-H.)

**Keywords:** cancer, glioblastoma, snoRNA-jouvence, acute myeloid leukemia, sh-lentivirus, BAALC, cell cycle

## Abstract

**Background/Objectives**: Cancer research aims to understand the cellular and molecular mechanisms involved, in order to identify new therapeutic targets and provide patients with more effective therapies that generate fewer side undesirable and toxic effects. Previous studies have demonstrated the role of small nucleolar RNAs (snoRNAs) in many physiological and pathological cellular processes, including cancers. SnoRNAs are a group of non-coding RNAs involved in different post-transcriptional modifications of ribosomal RNAs. Recently, we identified a new snoRNA (jouvence), first in Drosophila, and thereafter, by homology, in humans. **Methods**: Here, we characterize the effect of the knockdown of jouvence by a sh-lentivirus on human primary patient-derived glioblastoma cells. **Results**: The sh-lentivirus anti-jouvence induces a significant decrease in cell proliferation and leads to cell death. EdU staining confirmed this decrease, while TUNEL also showed the presence of apoptotic cells. An RNA-Seq analysis revealed a decrease, in particular, in the level of BAALC, a gene known to potentiate the oncogenic ERK pathway and deregulating p21, leading to cell cycle blockage. **Conclusions**: Altogether, these results allow the hypothesis that the knockdown of jouvence could potentially be used as a new anti-cancer treatment (sno-Therapy), especially against glioblastoma and also, potentially, against acute myeloid leukemia (AML) due to the BAALC deregulation.

## 1. Introduction

Glioblastoma is the most common primary malignant brain tumour, with a worldwide incidence of 10 per 100,000 members of the population [1]. Despite advances in research in surgery, radiotherapy, and chemotherapy improving the survival and quality of life of patients, this disease remains, in most cases, incurable, with a survival rate of only 14 months after diagnosis [1]. Patients have a high rate of recurrence and resistance to treatment. These figures therefore make it a crucial public health issue.

Recently, studies have demonstrated the role of small nucleolar RNAs (snoRNAs) in many physiological and pathological cellular processes, including cancers. Indeed, some snoRNAs show aberrant expression in several types of cancers and can also be used, for some of them, as markers of disease progression [2,3,4,5]. They are also deregulated between different stages and different types of cancer [6,7,8]. SnoRNAs are a group of non-coding RNAs contained in the nucleolus, the largest structure in the nucleus of eukaryotic cells involved in the synthesis of ribosomal RNAs (rRNAs), as well as the assembly and maturation of ribosomes [9]. SnoRNAs associate with proteins to form a complex, called small nucleolar ribonucleoprotein, carrying out different post-transcriptional modifications of rRNAs. The snoRNAs are divided into two major classes: C/D box snoRNAs are responsible for 2-O-methylation, while H/ACA box snoRNAs are responsible for the pseudouridylation of target nucleotides. Pseudouridylation, corresponding to the conversion of a uridine into pseudouridine, takes place at the level of the internal loops of snoRNAs, called pseudouridylation pockets [9]. Pseudouridine is one of the most abundant post-transcriptional modifications of RNA nucleotides [10]. As is well known, the role of ribosomes in protein translation is essential for cell growth and proliferation [11,12,13]. However, the excessive and uncontrolled proliferation of cancer cells leads to the formation of malignant tumors. It is therefore accepted that the involvement of snoRNAs in the modification of rRNAs could contribute to the progression of cancers [7,8].

Our recent research has demonstrated the involvement of a new snoRNA, identified in Drosophila, called snoRNA-jouvence (snoRNA-jou), in the process of longevity and protection against neurodegeneration [14,15]. Since snoRNAs are highly conserved during evolution [9], their counterparts in mice and humans have been identified, as they were not yet annotated in the genome [14,16]. The human genome has a single copy of snoRNA-jou of 159 base pairs, located on chromosome 11, in an intron of the TEAD1 gene [14,16]. The TEAD family of transcription factors plays different roles in cell proliferation, as well as tissue regeneration and homeostasis [17]. Structurally, human snoRNA-jou belongs to the class of H/ACA box snoRNAs [16]. However, it has a non-canonical secondary structure, i.e., it does not exhibit a classical “double hairpin” structure like most H/ACA box snoRNAs. However, its structure does not seem to affect its function, since a pseudouridylation pocket has been identified that can likely pseudo-uridylate 18S rRNA at position 1397 [16]. In order to characterize the role of snoRNA-jou in humans, first, its expression was studied on nine human cell lines, including immortalized cell lines. Three normal (non-cancerous) cell lines were studied, HEK (embryonic kidney), MRC5 (embryonic lung), and RPE1 (retinal pigment epithelium), as well as six cancerous cell lines: HCT116 and Caco2 (colon cancer), HeLa (cervical cancer), PC3 (prostate cancer), U87 (glioblastoma), and A549 (lung cancer). In all these cell types, the snoRNA-jou is detected at different levels, indicating that the snoRNA-jou is expressed in many cell types [16]. Studies were then carried out to study the overexpression and inactivation (knockdown) of snoRNA-jou in some of these lines. Overexpression of snoRNA-jou induces an increase in proliferation in all the studied cell types, while knockdown induces a decrease in cell proliferation [16]. In addition, a transcriptomic analysis (RNA-Seq) performed on the HCT116 cells overexpressing snoRNA-jou indicates the deregulation of several genes, while KEGG and GO analyses have suggested a genomic signature of a de-differentiation process, compatible with rejuvenation. Inversely, again on the HCT116 cells, an RNA-Seq performed on the knockdown of snoRNA-jou reveals a drastic decrease in the majority of the genes encoding for ribosomal proteins, indicating a collapse of ribogenesis, as well as several genes involved in the splicesome [16]. These two last effects of the knockdown correlate perfectly with the important decrease in cell proliferation phenotype.

In the studies carried out so far, the snoRNA-jou knockdown experiments were carried out using small interfering RNAs (siRNAs) on immortalized cell lines [16]. However, the transfection of cells by siRNAs is transient. This means that the DNA is not incorporated into the genome of the cells and, therefore, the expression of the siRNAs is no longer present in the daughter cells after cell division. Here, we decided to induce snoRNA-jou knockdown using lentivirus containing short hairpin RNAs (shRNAs) directed against the snoRNA-jou. The transduction of shRNAs by a viral vector makes it possible to more easily intervene in primary cells, which are usually more difficult to transfect by other techniques. Lentiviruses are RNA viruses that can integrate their vector into the genome of the host cell and therefore allow stable expression of the transgene, in this case, the shRNA. We decided to begin the experiments, first, on the HCT116, which was well characterized in our previous study [16], with the aim of validating the sh-lentivirus approach, and, second, on primary cells, the GBM14-CHA cells (xenograft), derived from a patient with glioblastoma. The obtained positive results suggest that the knockdown of snoRNA-jou could make it possible to block the proliferation of cancerous cells and, therefore, that it could potentially represent a new therapeutic tool against certain cancers.

## 2. Results

### 2.1. Knockdown of the snoRNA-Jou by Sh-Lentivirus Decreases Cancerous Cell Proliferation

To study the snoRNA-jou knockdown’s effects, human cancerous cells were transduced with a newly designed sh-lentivirus (Appendix A) to induce snoRNA-jou degradation using shRNA interference. We used different amounts of lentivirus to transduce the cells, corresponding to the multiplicity of infection (MOI). We first chose to study the effects on carcinoma cells (HCT116), as in our previous studies [16], in order to check whether siRNA-like effects could be observed with this new sh-lentivirus. Second, to study the effects on primary cells, we chose glioblastoma primary cells (GBM14). The proliferation of transduced cells was compared to that of non-transduced cells, to that of cells after the application of polybren, and to a sh-scramble-lentivirus. Cell proliferation (viability) measurements were performed by two independent methods. The first was the CellTiter-Glo assay, and the second was direct counting by Malassez cells, at different times after transduction, at 3, 8, and 10 days, to observe the longer-term effects of shRNAs after the division of transduced cells.

First, to determine the efficacy of the sh-lentivirus, we examined the dose–response effect of the cell proliferation (viability) on the HCT116 cells measured at 3 days post-treatment (Appendix A). The two controls, the polybren and the sh-control [mock (sh-scramble) lentivirus at MOI-20] did not generate any effects. However, we observed a decrease in cell proliferation by increasing the amount of sh-lentivirus anti-jouvence (MOI) used to transduce the cells. With a MOI-1, the cell proliferation is reduced by 10%, with MOI-5 by 20%, MOI-10 by 30%, MOI-20 by ~45%, and, finally, MOI-50 by 60%. Then, we performed all the following experiments with a MOI-20, which will be compared to the sh-scramble control.

More specifically, with a MOI-20, at 3 days post-transduction, HCT116 shows a decrease of 43% compared to the sh-scramble, an effect confirmed by the Malassez counts (50% decrease) (Figure 1a,b). In parallel, to check molecularly the efficacy of the sh-lentivirus, we carried out RT-qPCR (TaqMan) in order to quantify the level of expression of snoRNA-jou in cells transduced by sh-lentivirus. With a MOI-20, at 3 days, we found that the transduced HCT116 cells underexpressed snoRNA-jou by 49% (fold change ~0.5) compared to the non-transduced cells (Figure 1c). To verify that the observed effects were not specific only to HCT116 cells, we transduced other immortalized cell types with sh-lentivirus. After 3 days with the MOI-20, we observed similar results on lung cancer cell line A549, with a reduced viability of 52% (Figure 1d), and on glioblastoma cell line U87-MG, with a reduced viability of 57% (Figure 1e).

In a second step, as mentioned above, we investigated the sh-lentivirus on glioblastoma primary cells (GBM14). After 3 days, with the MOI-20, we observed a decrease in proliferation of 52%, as measured by CellTiter-Glo (Figure 1f), with a similar result (55%) when measured by direct Malassez cell count (Figure 1g). After 8 days, we observed a decrease of 71% (Figure 1h), and up to 84% at 10 days (Figure 1i). As with the HCT116, we checked at the molecular level, by RT-qPCR (TaqMan), the efficacy of the sh-lentivirus. With a MOI-20, we found that transduced GBM14 cells underexpressed snoRNA-jou by 60% (Figure 1j). Obviously, in both cases, we observed a partial (not total) knockdown of the amount of snoRNA-jou. However, this partial inhibition is sufficient to lead to a clear phenotype, a striking effect on cell proliferation. We also checked, on the primary cancerous cells (GBM14), as we did for the laboratory cell lines [16], whether our previous siRNA (LNA type) was also efficient in decreasing the cell proliferation. Indeed, Figure 1k shows that the proliferation is reduced by 26% after 3 days, which also roughly corresponds to a reduction of about 20% in the amount of the snoRNA-jouvence (Figure 1l). The difference in the percentage of the reduction in cell proliferation observed between the siRNA and the sh-lentivirus is likely due to the amount of siRNA entering into the cells, which are not integrated in the genome, therefore leading to a more transient effect, compared to sh-lentivirus. Finally, the visual microscopic observations concord with the cell proliferation analysis. Indeed, after 7 days post-transduction, the sh-scramble/control GBM14 showed much more cells than the sh-lentivirus treated cells (Figure 1m). In addition, since the lentivirus used contains the eGFP, we took advantage of this property to evaluate the transduction efficiency, which is about 100% (Appendix A). Altogether, these results confirm the efficacy of our new sh-lentivirus for the knockdown of the snoRNA-jou. Overall, these results therefore indicate that the knockdown of snoRNA-jou by sh-lentivirus induces a significant decrease in the proliferation of transduced cancer cells, with a dose–response effect, and with greater effects over time.

### 2.2. EdU Confirms the Decrease in Cell Proliferation

On the cell cultures directly observed under the inverted microscope, several cells looked darker, suggesting that they were affected (or in a dying process), for both HCT116 and GBM14 (Appendix A). Therefore, we characterized more precisely these effects. First, to corroborate the decrease in cell proliferation revealed by the CellTiter-Glo and the direct determination of the number of cells by the Malassez count, we performed EDU staining, which labels cells in division. As expected, for a MOI-20 at 3 days, the number of labeled cells (in proliferation) treated with the sh-jou-lentivirus was reduced by about 30% compared to the cells treated with the sh-scramble lentivirus (Figure 2a,b).

### 2.3. TUNEL Reveals the Presence of Apoptotic Cells

Second, in a further step, to assess whether the sh-jou-lentivirus would also lead to the death of the cells, we performed a TUNEL assay. For the MOI-20 at 3 days, we remark that several cells were already in apoptosis (Figure 3a,b), indicating that in addition to the blockage of cell proliferation, several cells entered apoptosis and, consequently, died.

### 2.4. Transcriptomic Analysis Reveals the Decrease in Several Genes, Including BAALC

In order to elucidate the molecular mechanisms responsible for the decrease in cell proliferation and cell death, we performed a transcriptomic analysis (RNA-Seq) by comparing the GBM14 cells treated with sh-jou-lentivirus (MOI-20, 3 days) with their appropriate control, the GBM14 cells treated with the sh-scramble (MOI-20). First, a Venn diagram revealed that 726 expressed genes were specific to the sh-lentivirus-treated cells, while 433 were specific to the sh-scramble control (Figure 4) (Appendix A). Among them, XIST, the X inactive specific transcript, as well as several genes of the histone cluster family, are specifically expressed in sh-jou, while TP63 or cell division cycle 25A (CDC25A) are specific to the sh-scramble controls, to name but a few. According to the standard criteria of a log2-fold change, and a *p*-value < 0.05, the differentially expressed genes (DEGs) analysis reveals that 666 genes are downregulated, while 825 genes are upregulated (Figure 5a) (see the Appendix A for the complete list of DEGs). In terms of the *p*-value (padj), the most deregulated gene is BAALC (Brain And Acute Leukemia, Cytoplasmic) [18], a gene responsible for acute myeloid leukemia (AML), and known to potentiate the oncogenic ERK pathway through interaction with MEKK1 and KLF4 [19]. In AML cell lines, it has been shown that high BAALC increases proliferation and accelerates the cell-cycle progression, while in contrast, its knockdown showed the opposite phenotype, meaning decreasing the cell cycle progression and/or blocking it [19]. Thus, we checked whether the reported genes interacting with BAALC were also deregulated in the GBM14 cells treated with sh-jou. Indeed, several ribosomal protein S6 kinases, such as RPS6KA1, A3, A4, and B1, the direct downstream targets of the ERK pathway, are deregulated (Table 1), as well as several mitogen-activated protein kinases, such as MAPK1 and MAP3K2, K4, K8, K9, K14, and K21 (Table 1), and Cyclin-dependent kinase 2 interacting protein (CINP) (Table 2). Moreover, as also formerly reported to interact with BAALC, the Oxysterol binding protein-like 1A (OSBPL1A) is deregulated [19]. Then, to move further, we checked for other OSBPL, and found that OSBPL5, 6, and 9 are also deregulated (Table 2). In a more general overview, we observed that up to 14 genes involved in cholesterol homeostasis are also deregulated (Table 2), including the 7-dehydrocholesterol reductase, or the Niemann–Pick (NPC1-like intracellular cholesterol transporter 1). Finally, several (up to 20) cyclin-dependent kinases (CDK) and various cyclins are also deregulated (Table 2), including the CDKN1A (p21), a well-known gene involved in cell cycle control. To support and validate that the deregulation of certain mRNAs also leads to a deregulation of their corresponding proteins, some Western blots were performed on HCT116 cells (Appendix A). As expected, p21 protein is increased, while the Cyclin B1 is decreased.

In complement, and to temptingly group together these various and numerous deregulated genes, the Gene Ontology (GO) analysis reveals that several pathways are deregulated, with the extracellular matrix and structure organization being the most deregulated (Figure 5b) (see Appendix A for the complete GO-enrichment). Among them, the ERK1 and ERK2 cascade (GO:0070371), the oncogenic pathway potentiated by high-BAALC, is significantly downregulated in GBM14-treated cells (see Table 1 for the list of involved genes), in accordance with the decrease in the level of expression of the BAALC gene observed here. Moreover, several other pathways affecting the sterol or the cholesterol are deregulated, such as the regulation of cholesterol efflux (GO:0010874, GO:0010875, GO:0033344), the regulation of sterol and/or cholesterol transport (GO:0032371, GO:0032374, GO:0032373, GO:0032376), the regulation of cholesterol storage (GO:0010885, GO:0010878), as well as lipid storage (GO:0019915, GO:0010883). Interestingly, the regulation of apoptotic signaling pathways (GO:2001233, GO:2001236) is also deregulated, as well as some pathways affecting the chromatin structure (GO:0006342, GO:0000183). However, the KEGG analysis did not reveal striking differences, only some pathways being statistically downregulated (Appendix A), while only five pathways are up-regulated (Appendix A).

### 2.5. Lipidomic Analysis Reveals a Deregulation of Cholesterol and Other Lipids

Cholesterol homeostasis has been implicated in several cellular and molecular mechanisms, and notably in cancer [20,21], while some other snoRNAs have already been involved in cholesterol metabolism [22]. As revealed by the transcriptomic analysis, several genes involved in cholesterol pathways are deregulated (Table 2), and as suggested by the GO analysis (Appendix A), are involved in either cholesterol efflux, transport, storage, or homeostasis. Thus, as a first step in the analysis, to investigate whether the deregulation of these several genes leads to a cellular phenotype, and notably whether cholesterol is indeed deregulated, as well as other lipids, we performed a lipidomic analysis on GBM14 cells treated with the sh-lentivirus anti-jouvence compared to the sh-scramble control. With a MOI-20 and 3 days post-transduction, we observed a strong increase in various forms of cholesterol, such as cholesterol itself, dihydrocholesterol, and cholesterol esters (Figure 6a). We also observed an increase in total lipids, triglycerides, and total phospholipids (Figure 6b). However, the precise determination of the composition of the fatty-acids (FAs) of these various classes of lipids did not reveal striking differences between the two conditions (sh-scramble-control versus sh-lentivirus anti-jou) (Appendix A). Indeed, overall, in all the lipid classes, each FA increased by a factor of about three in the sh-lentivirus anti-jou-treated cells compared to their controls, roughly similar to the increase observed for the corresponding lipid class itself. Altogether, these results indicate that the knockdown of jouvence perturbs quantitatively overall and general lipid and cholesterol homeostasis, without affecting specifically a precise FA pathway.

## 3. Discussion

This study validates the use of lentiviruses containing shRNAs as a tool for the inhibition (knockdown) of snoRNA-jou, both on three immortalized cell lines, and, more importantly, on patient-derived primary (glioblastoma) cells (xenograft), a terrifying cancer known to be refractory to almost all types of treatment. Indeed, on the colon carcinoma cell line (HCT116), the lung cancer cell line A549, and on the glioblastoma cell line U87-MG, as well as on primary glioblastoma cells (GBM14), transduction by sh-lentiviruses induces a significant decrease in cell proliferation and, as demonstrated in HCT116 cells, one that is proportional to the amount of lentivirus used (from MOI-1 up to MOI-50) (Appendix A). The integration of shRNA-jou into the cell genome seems to be functionally validated by the analysis conducted over longer periods. Indeed, cell proliferation decreases more significantly over time, demonstrating long-term effects and, therefore, the probable presence of shRNA-jou in the daughter cells after the division of the transduced cells. This result is also supported by si-RNA transfection, since the observed effect was weaker in this latter approach. In addition, the downregulation of snoRNA-jou was demonstrated by RT-qPCR in the treated cells, thus validating the use of sh-lentivirus as a knockdown tool for the snoRNA-jou. Finally, as revealed by the RNA-Seq analysis, the knockdown of snoRNA-jou induces significant deregulation of several genes, particularly some key genes, such as CDKN1A (p21), which is involved in cell cycle regulation (see also the Appendix A for the protein analysis by Western blot). Interestingly, in the GBM14 cells, the Venn Diagram (Figure 4) revealed that 433 genes were expressed only in the control line, meaning that their expression was totally extinct in the sh-jou treated cells. On the other hand, 726 genes, among them XIST, the X inactive specific transcript, as well as several genes of the “histone cluster family member” (Appendix A), all of which are known to modify the chromatin structure, were expressed de novo in the sh-jou-treated cells, while they were not expressed in the sh-scramble-treated control cells. Interestingly, a potential role for H/ACA snoRNAs in chromatin remodeling has already been reported [23], while in Drosophila, a chromatin-binding protein (Df31), which has been shown to play a role in maintaining open chromatin structure, is associated with C/D box snoRNAs [24]. Thus, this putative effect of snoRNA on chromatin could perhaps explain why such a snoRNA as jouvence could modulate the mRNA level of such a high number (1492) of genes.

The p21 protein, encoded by the CDKN1A gene, is an inhibitor of the cyclin E/cdk2 complex, thus disrupting the checkpoint of the G1/S transition (Figure 7) and inducing a blockage of the cell cycle in the G1 phase [25]. Knockdown of snoRNA-jou induces overexpression of the p21 protein, also supporting, here, the hypothesis of a cell cycle blockage, probably in the G1 phase. The TP53, a transcription factor, also well-known as a tumor suppressor gene, has been involved in many cancers [26]. Although the p53 itself is not deregulated in GBM14 sh-lentivirus anti-jouvence-treated cells compared to their control cells, at least seven genes known to be dependent on or interacting with the p53 are deregulated. Indeed, MDM4, TP53INP2 (tumor protein p53 inducible nuclear protein 2), TP53I13 (tumor protein p53 inducible protein 13), TP53RK (TP53 regulating kinase), TP53INP1 (tumor protein p53 inducible nuclear protein 1), and TP53I11 (tumor protein p53 inducible protein 11) are deregulated (Table 2), indicating that the perturbation of these genes might be likely to contribute to an effect on cell proliferation and/or the cell cycle.

Cholesterol homeostasis has been implicated in several cellular and molecular mechanisms in the pathophysiology of certain cancers [20,27] and/or associated with apoptosis [28,29,30]. Here, in sh-lentivirus anti-jou-treated GBM14 cells, several genes involved either in sterol or in cholesterol homeostasis are deregulated (Table 2), which correlates with elevated levels of different forms of cholesterol (cholesterol, dihydrocholesterol, and cholesterol esters), as determined by the lipidomics analysis. Notably, one of them, the Oxysterol binding protein-like 1A (OSBPL1A), which is also deregulated here, has already been reported to interact directly with BAALC [19], while a relationship between the p53 mutation and the mevalonate pathway (the main part of the cholesterol pathway) has been observed in mammary tissue [31]. Obviously, as already underlined by other studies, a relationship between the cholesterol level, apoptosis, and the cell cycle was also observed here, while this causal relationship remains to be thoroughly determined.

To conclude, we have demonstrated that the knockdown of a small non-coding RNA, the snoRNA-jouvence, has a significant anti-proliferative effect on some cancer cell lines, including patient-derived primary cells, as in glioblastoma. This effect seems primarily to be the consequence of the decrease in the abnormally high level of the BAALC gene, which has been shown to be enriched in glioma cells (The Human Protein Atlas: https://www.proteinatlas.org/ and [32]). The reduction in BAALC, in turn, might block the oncogenic ERK pathway and, consequently, lead to the blockage of several mitogen-activated protein kinases (MAP-kinases), yielding to the deregulation of the cell cycle, as also suggested here by a decrease in the EdU labeling. Interestingly, we also noticed that the reduction in BAACL gene expression is correlated with an increase in p21 mRNA, as reported in Acute Myeloid Leukemic (AML) cells. Thus, our study highlights the role of BAALC in glioblastoma, making it a potential target in cancer treatment. Chemotherapy drugs are designed to inhibit different mechanisms of cancer cells, such as DNA replication, protein synthesis, or tumor angiogenesis. Recent studies suggest that chemotherapy combined with cell cycle regulation strategies have positive effects for cancer treatment [33]. Although snoRNAs do not code for proteins, they have demonstrated an important involvement in many cellular processes, such as cell cycle regulation. Increasing studies are currently pointing to small nucleolar RNAs as promising potential biomarkers of cancer, either for detection or for the prognostic signature [2,34,35]. Therefore, targeting non-coding RNAs, such as the snoRNA-jouvence, represents an original and promising new strategy for cancer therapy, particularly against glioblastoma and likely against acute myeloid leukemia (AML).

## 4. Methods

### 4.1. Cell Lines and Culture Conditions

#### 4.1.1. Immortalized Cell Lines

HCT116 (human colon carcinoma), A549 (human lung carcinoma), and U87-MG (human glioblastoma) cell lines come from the American cell bank ATCC (American Type Culture Collection (Manassas, VA, USA).

#### 4.1.2. GBM14-CHA Primary Cells

The patient-derived primary GBM14-CHA cells (abbreviated as GBM14) come from the company XenTech (Evry-Courcouronnes, France), which specializes in the production of PDX (Patient-Derived Xenograft) cells. PDX cells are human tumor cells grafted into immunodeficient mice in order to be amplified, then dissociated and cultured. In this study, these are human glioblastoma cells taken from a 70-year-old woman.

#### 4.1.3. Culture Conditions

Cells are cultured according to standard protocols. HCT116 cells are cultured in McCoy’s 5A medium (Gibco™, ThermoFisher Scientific, Les Ulis, France) supplemented with 10% fetal calf serum (Gibco™). A549 cells are cultured in RPM1 medium (Gibco™) supplemented with 10% fetal calf serum (Gibco™). U87-MG cells are cultured in DMEM + GlutaMAX medium (Gibco™) supplemented with 10% fetal calf serum (Gibco™). GBM14 cells are cultured in DMEM/F-12 + GlutaMAX medium (Gibco™) supplemented with 8% fetal calf serum (Gibco™) and 1% penicillin–streptomycin (Gibco™). Cells are incubated at 37 °C and 5% CO_2_.

### 4.2. Lentivirus Design and Transduction

The lentivirus was designed by us, and produced by VectorBuilder™ (Chicago, IL, USA). Briefly, the sh-jou (hairpin) is placed under the control of the U6-promoter. The vector also contains the following markers: EGFP:T2A:Puromycine (Appendix A). These allow the expression of a short hairpin RNA (shRNA) targeting the sequence of snoRNA-jou (GGCTATTGTGGACAGAGGA) or a negative (CCTAAGGTTAAGTCGCCCTCG) control sequence (sh-scrambled). The viral particle titer of the ultra-purified lentivirus tubes is 3.10^9^ TU/mL. On the first day, the cells are counted, seeded, and incubated for 24 h. The next day, the lentivirus are applied to the cells in different quantities (MOI). Mixes containing medium, polybrene (increases transduction efficiency), and lentivirus are made according to the multiplicity of infection (MOI) determined. The MOI corresponds to the ratio between the number of viral particles used to infect the cells and the number of cells. The culture medium in which the cells are seeded is removed, and then the mixes containing the lentivirus are added to each well. The following day, the medium containing the lentivirus is removed and replaced with fresh medium. The control condition is cells treated with control “scramble” lentivirus (shRNA without target in the cell), abbreviated as “co”.

### 4.3. CellTiter-Glo Cell Viability Assay

The CellTiter-Glo test (Promega™, Charbonnières-les-Bains, France) measures cell viability, based on the quantification of ATP, a reliable indicator of metabolically active cells. The results obtained correspond to a quantity of luminescence, considered to be proportional to the number of living cells. The measurements are carried out on 96-well plates with a white background. CellTiter-Glo reagent is added directly to the wells containing the cells. The plate is shaken away from light (15 min), then the luminescence is measured with the GloMax Navigator plate reader (Promega™). For a one-plate read, each condition is represented by 10 wells. “Blank” wells with medium without cells are also measured. The analysis of the results consists of subtracting the average of the blanks from each raw datum, then dividing these values by the average of the control condition (sh-scramble treated cells) and multiplying them by 100. The average of each condition is thus calculated to obtain a percentage of relative luminescence, compared to the control condition. HCT116, A549, and U87-MG cells are seeded at 3000 cells and GBM14 cells are seeded at 5000 cells per well.

### 4.4. Cell Counting

Cells are counted using standard Malassez counting chamber, after staining with Trypan Blue (Promega™, Charbonnières-les-Bains, France) (6 µL of cells + 6 µL of Trypan Blue).

### 4.5. Transcriptomic Analysis (RNA-Seq)

The transcriptomic analysis (RNA-seq) was undertaken by Novogene (China, the Cambridge UK Branch), and performed as in El-Khoury et al. [16]. Briefly, total RNA from human cell lines were extracted using NucleoSpin RNA Plus columns (Macherey-Nagel, HŒRDT, France), according to the manufacturer’s instructions. Extracted RNAs were verified for the absence of genomic DNA contamination, by performing an RT-PCR using the ribosomal gene RP49. Contaminated samples were therefore treated with RQ1 DNase (Promega^TM^, Charbonnières-les-Bains, France) and cleaned with the NucleoSpin RNA Clean-UP (Macherey-Nagel, HŒRDT, France). After sample quality control, libraries enriched for polyA RNAs were generated and checked for quality. Then, the libraries were sequenced on an Illumina Hiseq platform and 125 bp/150 bp paired-end reads were generated. The resulting data were controlled and analyzed with bioinformatic tools. For all the computational tools analysis, common default parameter values were used. The reference genome and gene model annotation files were downloaded from genome website directly. Index of the reference genome was built using Bowtie v2.2.3 and paired-end clean reads were aligned to the reference genome using TopHat v2.0.12. We selected TopHat as the mapping tool because TopHat can generate a database of splice junctions based on the gene model annotation file and, thus, a better mapping result than other non-splice-mapping tools. For the quantification of gene expression level, HTSeq v0.6.1 was used to count the read numbers mapped to each gene. Then, FPKM of each gene was calculated based on the length of the gene and read counts mapped to this gene. FPKM, expected number of Fragments Per Kilobase of transcript sequence per Millions base pairs sequenced, considers the effect of sequencing depth and gene length for the read counts at the same time, and is currently the most commonly used method for estimating gene expression levels.

Differential expression analysis of two conditions (three biological replicates per condition) was performed using the DESeq R package (1.18.0). DESeq provides statistical routines for determining differential expression in digital gene expression data using a model based on the negative binomial distribution. The *p* values were adjusted using the Benjamini and Hochberg method. Corrected *p*-value of 0.05 and log2 (fold-change) of 1 were set as the threshold for significantly differential expression. The biological variation was eliminated (case with biological replicates), and the threshold was therefore normally set as *p* adjusted < 0.05. Gene Ontology (GO) enrichment analysis of differentially expressed genes was implemented by the GOseq R package, in which gene length bias was corrected. GO terms with corrected *p*-values less than 0.05 were considered significantly enriched by differential expressed genes. For the KEGG database [36,37], KOBAS software was used to test the statistical enrichment of differential expression genes in KEGG pathways.

### 4.6. Cell Proliferation ASSAY (EdU)

Proliferation tests were carried out using the Click-iT EdU assay with Alexa Fluor 594 (Invitrogen), according to the supplier’s instructions. Briefly, GBM14 cells were seeded in 8-well Lab-tek (ThermoFisher Scientific, Les Ulis, France), at 10,000 cells per well, and incubated overnight. The following day, the cells were treated with the lentivirus (control/scramble or sh-jou) at a MOI-20, and incubated for 3 days. Thereafter, the cells were incubated with the EdU solution for 5 h, to allow sufficient time to the proliferative cells to incorporate the EdU. Then, the cells were fixed with PFA 4%, during 15 min, washed twice with PBS with BSA 3%, and then incubated for 20 min in PBS containing Triton X-100 at 0.5%. Thereafter, the EdU detection was performed according to the Click-it Kit instructions (ref.: Kit EdU Click-iT-Alexa Fluor 594: C10339, ThermoFisher Scientific, Les Ulis, France). 

### 4.7. Apoptotic Cells Detection (TUNEL)

Apoptotic cells detection was carried out using the One-step TUNEL In Situ Apoptosis Kit according to the supplier’s instructions (TUNEL kit Red AF594, ref. E-CK-A322–20) (Elabscience ^TM^, Houston, TX, USA). Briefly, GBM14 cells were seeded in 8-well Lab-tek (ThermoFisher Scientific, Les Ulis, France), at 10,000 cells per well, and incubated overnight. The following day, the cells were treated with the lentivirus (control/scramble or sh-jou) with a MOI-20, and incubated for 3 days. Then, the TUNEL experiments were performed according to the manufacturer’s instructions.

### 4.8. Lipidomic Analysis

Lipid analysis was performed on the Functional Lipidomics Platform acknowledged by IBiSA (Infrastructure in Biology, Health and Agronomy). After addition of appropriate internal standards (1,2-diheptadecanoyl-*sn*-glycero-3-phosphocholine, cholesteryl ester 17:0, tri-17:0 triglyceride, and ^13^C cholesterol,) to the samples, total lipids were extracted twice with chloroform/ethanol (2:1, *v*/*v*). The organic phases were dried under nitrogen and the different lipid classes were then separated by thin-layer chromatography using the solvent mixture hexane–diethylether–acetic acid (80:20:1, *v*/*v*/*v*) as eluent. Lipids were revealed by UV light after spraying the plate with 0.02% dichlorofluorescein in ethanol and identified by comparison with standards. Silica gel was scraped off. Total lipids, total phospholipids, triacylglycerols, and cholesteryl esters were transmethylated, and the fatty acid methylesters were analyzed by gas chromatography. Briefly, samples were treated with toluene–methanol (2:3, *v*/*v*) and 14% boron trifluoride in methanol. Transmethylation was carried out at 100 °C for 90 min. The reaction was stopped by cooling the samples to 0 °C and by the addition of 1.5 mL K_2_CO_3_ in 10% water. The resulting fatty acid methyl esters were extracted with 2 mL of isooctane and analyzed by gas chromatography using an HP6890 instrument equipped with a fused silica capillary BPX70 SGE column (60 × 0.25 mm), with hydrogen as a vector gas. Temperatures of the flame ionization detector and the split/splitless injector were set at 250 °C and 230 °C, respectively. Peak detection and integration were performed using ChemStation software (version LTS 01.11).

Cholesterol was extracted twice with chloroform/ethanol (2:1, *v*/*v*). The dry residue was derivatized with 100 μL N,O-bis-trimethylsilyl-trifluoroacetamide (BSTFA) for 20 min at 60 °C. Derivatized cholesterol was then analyzed by gas chromatography (HP 7890B, Agilent, Santa Clara, CA, USA) coupled with triple quad mass spectrometry (GC-MS/MS) using the electron impact ionization (EI) mode (7000C, Agilent). GC-MS/MS was equipped with a SolGel-1ms fused silica capillary column, 60 m length, 0.25 mm internal diameter (Trajan, SGE, ThermoFisher Scientific, Les Ulis, France). The carrier gas was helium with a constant flow rate of 1.2 mL/min. The split/splitless injector was heated to 280 °C. The oven temperature program started at 55 °C for 4 min and then ramped up to 250 °C at 40 °C/min, and finally up to 310 °C at 20 °C/min for 25 min. The samples were injected in a splitless mode. The temperature of the mass spectrometer transfer line was set at 250 °C, while the source temperature was set at 200 °C. Nitrogen was used as collision gas (1.5 mL/min). The electron ionization energy was 70 eV. Cholesterol was detected using the multiple-reaction monitoring (MRM) mode.

### 4.9. Statistical Analysis

Data were analyzed statistically using one-tailed unpaired *t*-test with GraphPad Prism software (version 5).

## Figures and Tables

**Figure 1 ncrna-11-00054-f001:**
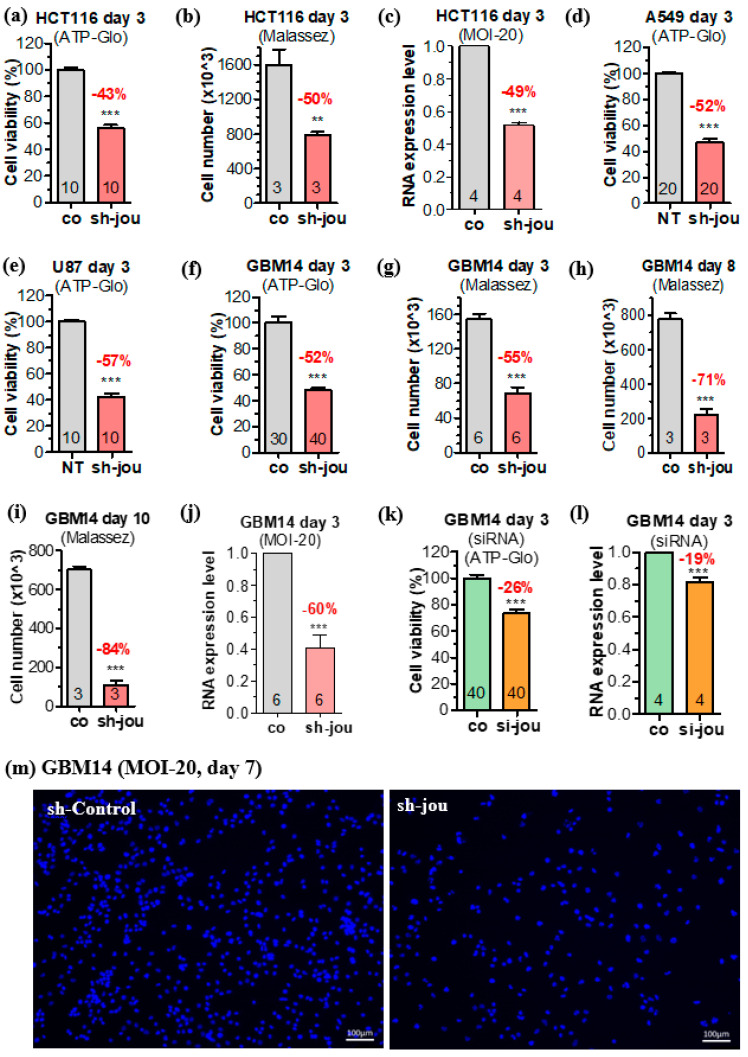
Knockdown of snoRNA-jou by sh-lentivirus decreases cancerous cell proliferation. (**a**,**b**) HCT116 cells’ proliferation at 3 days after transduction with a sh-lentivirus control without target (sh-co), versus cells transduced with sh-lentivirus targeting snoRNA-jou with a MOI-20. (**a**) Cell viability as determined by CellTiter-Glo. (**b**) Number of cells, as determined by Malassez cell counting. (**c**) The expression level of snoRNA-jou is decreased by about 50%, 3 days after transduction (MOI-20). (**d**) Lung cancer cell line A549 at 3 days, MOI-20, as determined by CellTiter-Glo. (**e**) Glioblastoma cell line U87-MG after 3 days, MOI-20, as determined by CellTiter-Glo. (**f**,**g**) GBM14 cells’ proliferation at 3 days transduced with a sh-lentivirus control without target (sh-co), versus cells transduced with sh-lentivirus targeting snoRNA-jou at a MOI-20. (**f**) Cell viability as determined by CellTiter-Glo. (**g**) Number of cells, as determined by Malassez cell counting. (**h**) As in G, but after 8 days post-transduction. (**i**) After 10 days post-transduction. (**j**) In GBM14, the expression level of snoRNA-jou is decreased by about 60%, 3 days after transduction (MOI-20). (**k**) GBM14 transduced with the siRNA, at 3 days post-transduction, as determined by the Cell-Titer-Glo. (**l**) In GBM14, the expression level of snoRNA-jou is decreased by about 20%, 3 days after transduction with siRNA. (**m**) Microphotography of the GBM14 cells stained with Hoechst, 7 days post-transduction, with sh-lentivirus compared to sh-scramble control. Statistics: (*p*-values: ** *p* < 0.001, *** *p* < 0.0005). Error bars represent the mean +/− SEM (*p*-values were calculated using the one-tail unpaired *t*-test using GraphPad Prism_5). Cell proliferation of each condition was compared to the sh-scramble (control) condition. For the CellTiter-Glo, numbers in each histogram represent the number of wells (e.g.,: 40 = 4 times 10 wells, and so forth), while for the Malassez counts, or the RNA expression level, they represent the numbers of replicates.

**Figure 2 ncrna-11-00054-f002:**
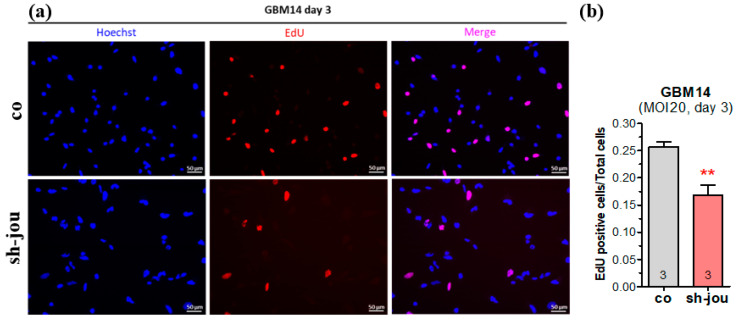
Knockdown of snoRNA-jou by sh-lentivirus reduces cell proliferation of GBM14, as revealed by EdU. (**a**) Microphotography of the GBM14 cells treated with sh-lentivirus compared to sh-scramble lentivirus (co), 3 days post-transdution (MOI-20). First column: Hoechst staining (in blue). Middle column: EdU staining (in red). Third column: Merge. (**b**) Histogram showing the number of labeled EdU-positive cells compared to total number of cells (*p*-values: ** *p* < 0.001).

**Figure 3 ncrna-11-00054-f003:**
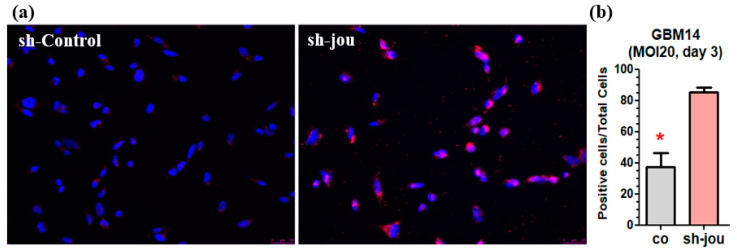
Knockdown of snoRNA-jou by sh-lentivirus induces cells death of GBM14, as revealed by TUNEL assay. (**a**) Microphotography of the GBM14 cells treated with sh-lentivirus compared to sh-scramble lentivirus (control) 3 days post-transduction (MOI-20). In red: TUNEL staining. In blue: Hoechst staining. In pink: overlay. (**b**) Histogram showing the number of labeled TUNEL-positive cells compared to total number of cells (*p*-values: * *p* < 0.05).

**Figure 4 ncrna-11-00054-f004:**
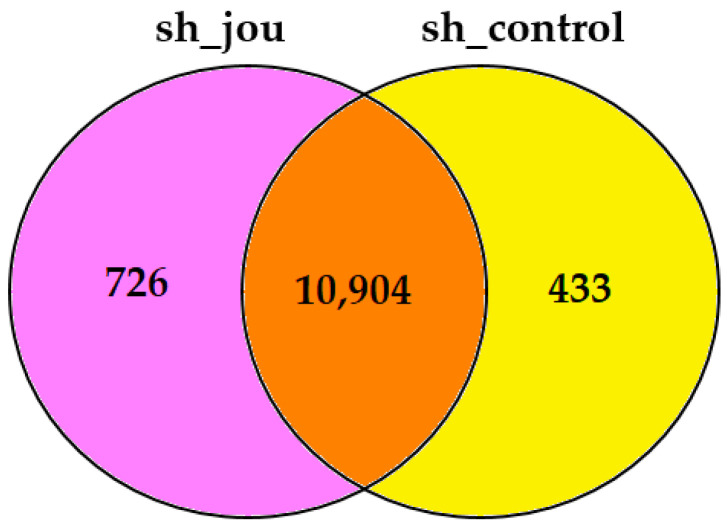
Venn diagram of the transcriptomic analysis comparing GBM14 sh-lentivirus-treated cells with control (sh-scramble lentivirus treated) GBM14. In total, 10,904 genes are expressed in common, while 433 are expressed only in control cells, and 726 only in sh-lentivirus-treated cells (see Appendix A for the complete list of genes).

**Figure 5 ncrna-11-00054-f005:**
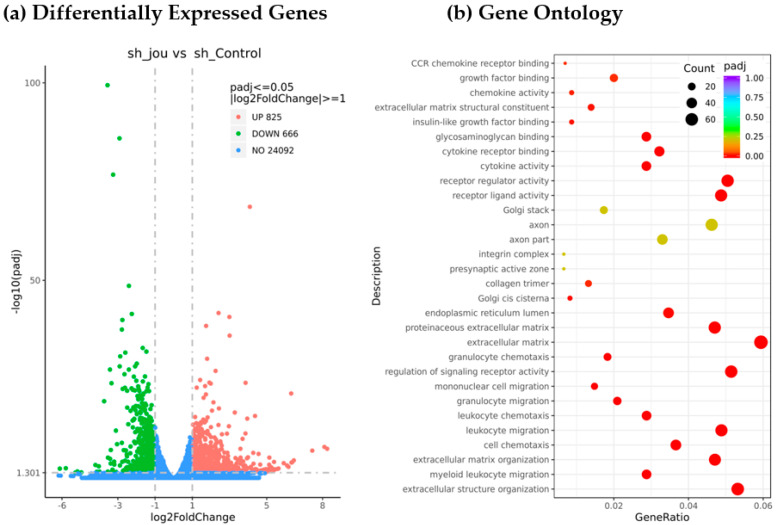
Transcriptomic analysis performed on total RNA from the knockdown of snoRNA-jou by sh-lentivirus on GBM14 cells compared to the control (sh-scramble lentivirus treated) cells. (**a**) Differentially Expressed Genes (DEGs) analysis reveals that 825 genes are upregulated, while 666 are downregulated (see Appendix A for the complete list of genes). (**b**) Gene Ontology analysis (GO) revealing several deregulated pathways (see Appendix A for the complete list of GO-enrichment).

**Figure 6 ncrna-11-00054-f006:**
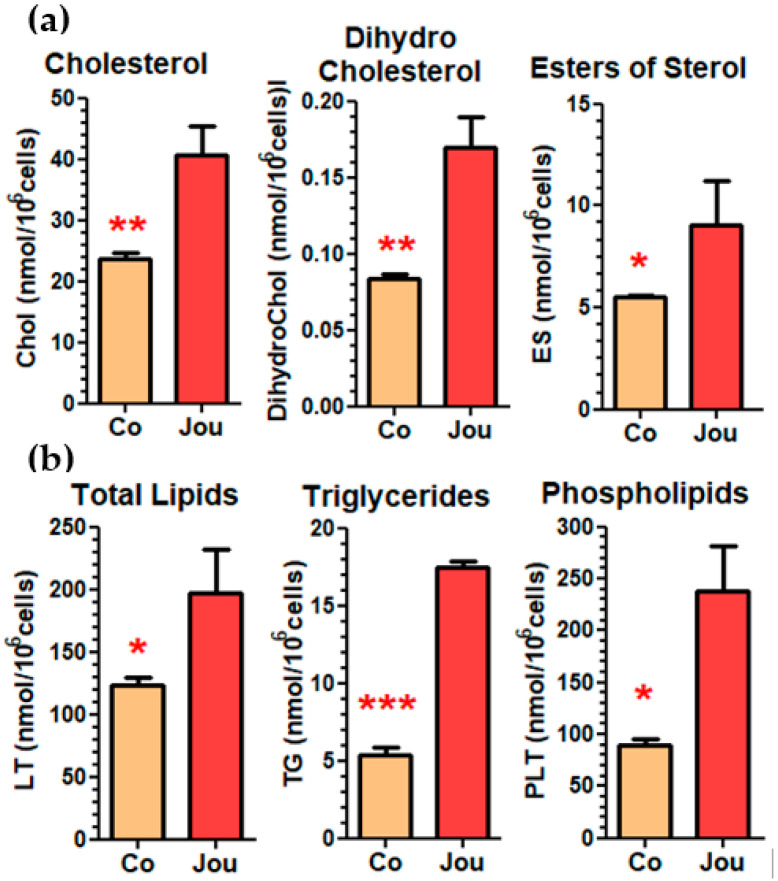
Lipidomic analysis reveals an important increase in different forms of cholesterol and lipids. Knockdown of snoRNA-jou by sh-lentivirus compared to sh-scramble lentivirus (control = co), 3 days post-transduction (MOI-20) induces an important increase in cholesterol, dehydrocholesterol, and esters of sterol (**a**), and in total lipids, triglycerides, and phosphilipids (**b**). Statistics: *p*-values: * *p* < 0.05; ** *p* < 0.005; *** *p* < 0.0005. Errors bars represent the mean +/− SEM (*p*-values were calculated using the one-tail unpaired *t*-test using GraphPad Prism) (*n* = 3 replicates).

**Figure 7 ncrna-11-00054-f007:**
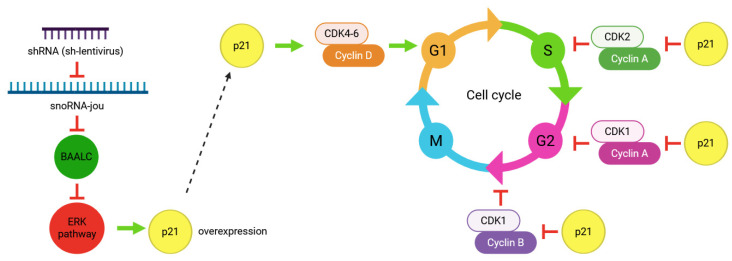
Working model depicting how the knockdown of the snoRNA-jou might affect the cell cycle. Knockdown of snoRNA-jou by sh-lentivirus decreases the level of the abnormally high-BAALC expressed gene in glioblastoma, which in turn inhibits ERK cascade by deregulating several MAP-kinases. These last, by a molecular mechanism that remains to be determined, increase p21, which acts at several steps of the cell cycle, stimulating the CDK4–6/cyclinD complex, inhibiting the cyclinA-CDK2 complex, thereby inhibiting the S phase. p21 is also known to inhibit the cyclinA-CDK1 complex, leading to the inhibition of the G2 phase. Finally, p21 also inhibits the cyclinB–CDK1 complex, which blocks the G2/M transition.

**Table 1 ncrna-11-00054-t001:** List of some selected deregulated genes (up or down) involved in the ERK1 and ERK2 cascade, in Mitogen-Activated Protein Kinase, and in Ribosomal Protein S6 Kinase.

Gene_ID	Fold Ch.	padj	Name	Gene_Description
**ERK1 and ERK2 cascade**				
ENSG00000164929	0.09	0.000	BAALC	BAALC, MAP3K1 and KLF4 binding
ENSG00000133048	0.37	0.000	CHI3L1	chitinase 3 like 1
ENSG00000100234	0.36	0.000	TIMP3	TIMP metallopeptidase inhibitor 3
ENSG00000204356	0.27	0.000	NELFE	negative elong, factor complex member E
ENSG00000162545	0.27	0.000	CAMK2N1	Ca^2+^/calmodulin dep. Prot. kinase II inhib.1
ENSG00000108691	0.23	0.000	CCL2	C-C motif chemokine ligand 2
ENSG00000115414	0.37	0.000	FN1	fibronectin 1
ENSG00000141736	0.32	0.000	ERBB2	erb-b2 receptor tyrosine kinase 2
ENSG00000090339	0.45	0.000	ICAM1	intercellular adhesion molecule 1
ENSG00000117228	0.39	0.000	GBP1	guanylate binding protein 1
ENSG00000006210	0.32	0.000	CX3CL1	C-X3-C motif chemokine ligand 1
ENSG00000152518	0.41	0.000	ZFP36L2	ZFP36 ring finger protein like 2
ENSG00000003402	0.41	0.000	CFLAR	CASP8 and FADD like apoptosis regulator
ENSG00000120875	0.49	0.000	DUSP4	dual specificity phosphatase 4
ENSG00000154188	0.44	0.000	ANGPT1	angiopoietin 1
ENSG00000136848	0.49	0.000	DAB2IP	DAB2 interacting protein
ENSG00000006606	0.40	0.000	CCL26	C-C motif chemokine ligand 26
ENSG00000113721	0.47	0.000	PDGFRB	platelet derived growth factor receptor b
ENSG00000108688	0.20	0.000	CCL7	C-C motif chemokine ligand 7
ENSG00000185650	0.41	0.000	ZFP36L1	ZFP36 ring finger protein like 1
ENSG00000169220	0.46	0.000	RGS14	regulator of G protein signaling 14
ENSG00000125845	0.48	0.000	BMP2	bone morphogenetic protein 2
ENSG00000148053	0.29	0.000	NTRK2	neurotrophic receptor tyrosine kinase 2
ENSG00000145242	0.40	0.001	EPHA5	EPH receptor A5
ENSG00000108700	0.06	0.001	CCL8	C-C motif chemokine ligand 8
ENSG00000019582	0.44	0.003	CD74	CD74 molecule
ENSG00000271503	0.10	0.009	CCL5	C-C motif chemokine ligand 5
ENSG00000183578	0.28	0.016	TNFAIP8L3	TNF alpha induced protein 8 like 3
**mitogen-activated protein kinase 11**				
ENSG00000006062	2.15	0.000	MAP3K14	mitogen-act. prot. kinase kin. Kin. 14
ENSG00000185386	0.54	0.012	MAPK11	mitogen-activated protein kinase 11
ENSG00000065559	1.66	0.000	MAP2K4	mitogen-activated protein kinase kinase 4
ENSG00000071054	0.71	0.001	MAP4K4	mitogen-act. prot. kinase kin. Kin. Kin. 4
ENSG00000100030	0.74	0.015	MAPK1	mitogen-activated protein kinase 1
ENSG00000109339	0.64	0.055	MAPK10	mitogen-activated protein kinase 10
ENSG00000138834	2.62	0.000	MAPK8IP3	mitogen-act. prot. kinase 8 interac. prot. 3
ENSG00000006062	2.15	0.000	MAP3K14	mitogen-act. prot. kinase kinase kinase 14
ENSG00000006432	1.77	0.000	MAP3K9	mitogen-act. prot. kinase kinase kinase 9
ENSG00000143674	2.04	0.002	MAP3K21	mitogen-act. prot. kinase kinase kinase 21
ENSG00000085511	1.33	0.019	MAP3K4	mitogen-act. prot. kinase kinase kinase 4
ENSG00000107968	0.66	0.050	MAP3K8	mitogen-act. prot. kinase kinase kinase 8
ENSG00000169967	1.27	0.066	MAP3K2	mitogen-act. prot. kinase kinase kinase 2
**ribosomal protein S6 kinase A1**				
ENSG00000117676	0.67	0.007	RPS6KA1	ribosomal protein S6 kinase A1
ENSG00000177189	0.74	0.015	RPS6KA3	ribosomal protein S6 kinase A3
ENSG00000162302	1.37	0.048	RPS6KA4	ribosomal protein S6 kinase A4
ENSG00000108443	1.38	0.040	RPS6KB1	ribosomal protein S6 kinase B1

**Table 2 ncrna-11-00054-t002:** List of some selected deregulated genes (up or down) involved in the Cell Cycle, Cyclins and CDK, in TP53 and associated proteins, and in cholesterol and sterol pathways.

Gene_ID	Fold Ch.	padj	Name	Gene_Description
**cell cycle, cyclin, and CDK**				
ENSG00000058091	2.33	0.000	CDK14	cyclin dependent kinase 14
ENSG00000108465	3.71	0.000	CDK5RAP3	CDK5 regulatory subunit ass. prot. 3
ENSG00000111328	0.63	0.001	CDK2AP1	cyclin dependent kinase 2 ass. prot. 1
ENSG00000138395	3.49	0.001	CDK15	cyclin dependent kinase 15
ENSG00000129355	0.57	0.002	CDKN2D	cyclin dependent kinase inhibitor 2D
ENSG00000164885	0.68	0.014	CDK5	cyclin dependent kinase 5
ENSG00000156345	0.61	0.017	CDK20	cyclin dependent kinase 20
ENSG00000100865	0.69	0.012	CINP	cyclin dependent kinase 2 inter. prot.
ENSG00000129757	0.63	0.023	CDKN1C	cyclin dependent kinase inhibitor 1C
ENSG00000124762	1.35	0.038	CDKN1A	cyclin dependent kinase inhibitor 1A
ENSG00000167258	1.38	0.061	CDK12	cyclin dependent kinase 12
ENSG00000185862	3.69	0.000	EVI2B	ecotropic viral integration site 2B
ENSG00000198625	1.74	0.001	MDM4	MDM4, p53 regulator
ENSG00000136807	1.49	0.004	CDK9	cyclin dependent kinase 9
ENSG00000123374	1.38	0.007	CDK2	cyclin dependent kinase 2
ENSG00000009950	2.54	0.061	MLXIPL	MLX interacting protein like
ENSG00000163660	2.62	0.000	CCNL1	cyclin L1
ENSG00000112576	0.58	0.000	CCND3	cyclin D3
ENSG00000221978	2.15	0.002	CCNL2	cyclin L2
ENSG00000147082	9.28	0.003	CCNB3	cyclin B3
ENSG00000118971	2.88	0.022	CCND2	cyclin D2
ENSG00000113328	0.74	0.024	CCNG1	cyclin G1
ENSG00000134480	1.41	0.041	CCNH	cyclin H
ENSG00000107443	1.56	0.042	CCNJ	cyclin J
**TP53 and associated protein**				
ENSG00000078804	0.41	0.000	TP53INP2	tumor prot. p53 inducible nuclear prot. 2
ENSG00000167543	0.55	0.009	TP53I13	tumor prot. p53 inducible protein 13
ENSG00000172315	0.65	0.034	TP53RK	TP53 regulating kinase
ENSG00000164938	1.64	0.069	TP53INP1	tumor protein p53 inducible nuclear prot. 1
ENSG00000175274	0.69	0.070	TP53I11	tumor protein p53 inducible protein 11
ENSG00000246640	4.46	0.001	PICART1	p53-inducible cancer-ass. RNA transcript 1
ENSG00000120279	4.03	0.002	MYCT1	MYC target 1
**Cholesterol and sterol pathways**				
ENSG00000116133	3.53	0.000	DHCR24	24-dehydrocholesterol reductase
ENSG00000138135	0.32	0.000	CH25H	cholesterol 25-hydroxylase
ENSG00000198911	0.40	0.000	SREBF2	sterol regul. element binding TF 2
ENSG00000160285	0.57	0.000	LSS	lanosterol synthase
ENSG00000072310	0.56	0.001	SREBF1	sterol regulatory element binding TF 1
ENSG00000109929	1.74	0.009	SC5D	sterol-C5-desaturase
ENSG00000079156	1.84	0.006	OSBPL6	oxysterol binding protein like 6
ENSG00000021762	1.43	0.014	OSBPL5	oxysterol binding protein like 5
ENSG00000141447	0.70	0.044	OSBPL1A	oxysterol binding protein like 1A
ENSG00000117859	1.26	0.068	OSBPL9	oxysterol binding protein like 9
ENSG00000015520	3.29	0.084	NPC1L1	NPC1 like intra. cholesterol transporter 1
ENSG00000172893	1.37	0.094	DHCR7	7-dehydrocholesterol reductase

## Data Availability

The authors declare that all the data and the methods used in this study are available within this article. The Appendix A are available from the corresponding authors upon reasonable request.

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
