# Peer review of "Knockdown of the snoRNA-Jouvence Blocks the Proliferation and Leads to the Death of Human Primary Glioblastoma Cells"

_ncrna, 2025, doi:10.3390/ncrna11040054_

Round 1

Reviewer 1 Report

Comments and Suggestions for Authors

The manuscript evaluates the effect of depleting the snoRNA-jouvence in cancer cell lines and especially a GBM cell line. The data on BAALC, ERK and p21 and lipids are rather clear on RNA / lipid level but remains unclear about how this is achieved. How does a snoRNA in nucleolus do all this? Though the data looks specific, it still relies on one short hairpin virus (and one control). It would have been good to see untransduced cells, and one additional hairpin, as well as a rescue experiment.

There is very little validation of the data like immunoblotting, PCR and functional assays. There is a risk to overinterpret the RNAseq findings (that could be a bit general effects).

The connection to the lipidomics is also not very clear, and what it means and how it is connected to snoRNA-jou.

The team use the GBM14 cells (PDX) that is good but it is only one more recently derived line that is tested so it is hard to draw more general conclusion, in addition it is cultured in serum (so one can as well use a panel of traditional glioma lines, like the U lines). To follow up on this there is lack of clinical correlation data e g snoRNA-jou, BAALC, so the rationale for looking into glioma, and the use (rationale) of GBM cells is thus not fully clear.

Can overexpression of BAALC or snoRNA-jouvence restore proliferation?

Structure of the paper, it is well written but can be made shorter, because there is a lot of repetition.

The RNAseq was done by company, and is referred to another reference, it would still be good to include basics such as biological replicates, comments on batch effects how they were controlled in RNA-seq or lipidomics. No discussion of RNA-seq alignment quality, read depth, or filtering criteria. Statistical analysis a bit unclear on test and whether they are one or two sided. I would suggest to include as much info on this RNAseq analysis and RNA isolation as possible in this paper. The reader should have this information and not go to company site or dig into the literature for the other references. Sometimes it is good to do so for examaple for IF and WB, but for RNAseq and the unique design there is for experiments I strongly suggest to include more info. Especially since the paper is not mechanistically clear. 

In summary, the referre understands that it is tricky to solve all mechanisms involved but the paper would benefit and be in better shape for publication if included more background and details on (1) RNAseq and validation of RNAseq data (necessary). (2)Improvement of figure presentation (can be made more clear), condensing the text a bit, try to connect and motivate the lipid data better (strongly recommended). (3)Some data mining could support the GBM data (much recommended). (4)Rescue experiments and additional tool to validate phenotype (recommended).  

Author Response

Reviewer-1)

The manuscript evaluates the effect of depleting the snoRNA-jouvence in cancer cell lines and especially a GBM cell line. The data on BAALC, ERK and p21 and lipids are rather clear on RNA / lipid level but remains unclear about how this is achieved. How does a snoRNA in nucleolus do all this? Though the data looks specific, it still relies on one short hairpin virus (and one control). It would have been good to see untransduced cells, and one additional hairpin, as well as a rescue experiment.

The controls have been performed with a “mock/scramble” sh-RNA, which is generally the rules for this kind of studies.

Moreover, for the Non-transduced cells (NT), they can be seen in Suppl. Figure 1, as well as one experiment with only Polybrene (PB) as well as with the sh-control (scramble) (at MOI-20). Sorry, but the Suppl. Figures data were not included (this was my mistake when I have submitted the manuscript).

For the rescue experiments, it is not obvious to “How” rescue a knock-down by a sh-RNA. Moreover, this kind of experiments can be very long and beyond the goal of this first study. 

There is very little validation of the data like immunoblotting, PCR and functional assays. There is a risk to overinterpret the RNAseq findings (that could be a bit general effects).

For a validation of the KD of jouvence, we have added a Western Blot on 2 different deregulated genes (p21 and Cyclin B1), demonstrating that the protein, as the mRNA levels, are deregulated (in Suppl. Figures 4).

To overcome the possible “general effect”  generally due to the inflammatory responses by the cells, the controls have been done with a “scramble sh-lentivirus, which by itself induces a “inflammatory effect” as any lentivirus infection. Thus this undesirable “general effect” can be ruled-out, since it is already included in the control. 

The connection to the lipidomics is also not very clear, and what it means and how it is connected to snoRNA-jou.

The connection between the lipidomics and the snoRNA-jou is based on the fact that several genes involved in lipids and cholesterol metabolism are deregulated in the KD of jouvence by sh-lentivirus anti-jouvence. Interestingly, this effect strongly remind the effect observed in another cell line, HCT116 studied and published earlier in El-Khoury et al., BMC-Genomics, 2020, as well as in Drosophila (Soulé and Martin, bioRxiv, 2020).

Thus, as a first step, to demonstrate that this deregulation of several metabolism genes leads to a real cellular effect, we decide to quantify the lipids and cholesterol through a lipidomic analysis. We consider that, in fine, this is the ultimate experiment to demonstrate that this deregulated genes lead to a biological effect.     

The team use the GBM14 cells (PDX) that is good but it is only one more recently derived line that is tested so it is hard to draw more general conclusion, in addition it is cultured in serum (so one can as well use a panel of traditional glioma lines, like the U lines). To follow up on this there is lack of clinical correlation data e g snoRNA-jou, BAALC, so the rationale for looking into glioma, and the use (rationale) of GBM cells is thus not fully clear.

We have volontarily choose the GBM14 line (a primary glioblastoma line), because after our first demonstration on the effect of KD of jouvence by si-RNA described and published in El-Khoury et al., BMC-Genomics, 2020 (Figure 4), performed on well established laboratory cells lines, we wonder in this effect could also be observed in primary cells line, as a step forward to go toward the development of a new potential therapeutic approach. Moreover, since primary cells lines are often more difficult to transduced by si-RNA, for which we have to use lipofectamin (or related products), we decide to develop a sh-lentivirus, which turned out to be more efficient, as expected. The choice of GBM14 (glioblastoma) was only based on the fact that this kind of terrifying cancer don’t have any real efficient treatment up to now, and thus we though that could be a good starting example. We are also aware that this could likely be efficient in other types of cancers (as mentioned at the end of the discussion).  

Can overexpression of BAALC or snoRNA-jouvence restore proliferation?

This is a very good comment and question, but this kind of experiments will be really long and as mentioned above, beyond the goal of this first study.

Structure of the paper, it is well written but can be made shorter, because there is a lot of repetition.

O.K., I have shorten the mansuscript in some places, and try to avoid as much as I can the repetition (see the modified text) labeled: (**text removed).

The RNAseq was done by company, and is referred to another reference, it would still be good to include basics such as biological replicates, comments on batch effects how they were controlled in RNA-seq or lipidomics. No discussion of RNA-seq alignment quality, read depth, or filtering criteria. Statistical analysis a bit unclear on test and whether they are one or two sided. I would suggest to include as much info on this RNAseq analysis and RNA isolation as possible in this paper. The reader should have this information and not go to company site or dig into the literature for the other references. Sometimes it is good to do so for examaple for IF and WB, but for RNAseq and the unique design there is for experiments I strongly suggest to include more info. Especially since the paper is not mechanistically clear. 

O.K., I have included much more information on RNA-Seq in the M&M.

In summary, the referre understands that it is tricky to solve all mechanisms involved but the paper would benefit and be in better shape for publication if included more background and details on (1) RNAseq and validation of RNAseq data (necessary). (2)Improvement of figure presentation (can be made more clear), condensing the text a bit, try to connect and motivate the lipid data better (strongly recommended). (3)Some data mining could support the GBM data (much recommended). (4)Rescue experiments and additional tool to validate phenotype (recommended).  

To resume, in brief:

1) I have included a detailed protocol in M&M, for the RNA-Seq  

2a) I have improve the presentation of the Figure 1 (also in accordance to the critics of the referee No 3). 

2b) To try to connect and motivate the lipid data, I have added the sentence:  “Thus, as a first step of analysis, to investigate if the deregulation of these several genes leads to a cellular phenotype, and notably if the cholesterol …… (lines 292-293)

3) Rescue experiements: we have not done tjhem, because we consider that it is beyond of the main goal of this first study. Morever, the allowed time (10 days for revision) do not allow such long and time consuming experiements, which notably, need to generate a “construction vector” to overexpress BAALC.

To validate in part the KD of jouvence, we have added a Figure (Supplementary Figure 4) showing the validation of the KD of jouvence at the level of the protein (on p21 and Cyclin B1), by Western Blots (done on HCT116 cells), demonstrating that indeed, the deregulation of genes (mRNA level) leads also to the deregulation of the corresponding proteins.

Few others modifications within the text, to improve the text:

     - change: lead to “contribute” (line 358)

     - in turn might block (line 385) I have added the word “ might” to be less affirmative, since we don’t have any direct demonstration of that  !

Reviewer 2 Report

Comments and Suggestions for Authors

The manuscript "Knockdown of the snoRNA-Jouvence Blocks Cell Proliferation and Leads to Cell Death of Human Primary Glioblastoma Cells" by Jaque-Cabrera and colleages presents a compelling investigation into the function of the small nucleolar RNA jouvence (snoRNA-jou) in glioblastoma biology. The authors demonstrate that shRNA-mediated knockdown of snoRNA-jou significantly impairs proliferation and induces apoptosis in both immortalized cancer cell lines and primary patient-derived glioblastoma cells. Using complementary assays (cell proliferation, EdU, TUNEL, RNA-seq, and lipidomics), the authors show that snoRNA-jou knockdown affects BAALC expression, ERK signaling, cell cycle regulators, and cholesterol homeostasis. Here are some comments:

1. The link between snoRNA-jou and BAALC remains correlative. It's unclear whether BAALC is directly regulated by snoRNA-jou or affected indirectly via global RNA processing or chromatin effects. Clarify whether the snoRNA binds or modulates BAALC mRNA directly or indirectly.

2.Figures 4 and 5a both compare sh_jou with shControl, but the total number of genes reported differs between them. Could you please clarify the reason for this discrepancy? It is currently unclear and somewhat confusing.

Author Response

Reviewer-2)

The manuscript "Knockdown of the snoRNA-Jouvence Blocks Cell Proliferation and Leads to Cell Death of Human Primary Glioblastoma Cells" by Jaque-Cabrera and colleages presents a compelling investigation into the function of the small nucleolar RNA jouvence (snoRNA-jou) in glioblastoma biology. The authors demonstrate that shRNA-mediated knockdown of snoRNA-jou significantly impairs proliferation and induces apoptosis in both immortalized cancer cell lines and primary patient-derived glioblastoma cells. Using complementary assays (cell proliferation, EdU, TUNEL, RNA-seq, and lipidomics), the authors show that snoRNA-jou knockdown affects BAALC expression, ERK signaling, cell cycle regulators, and cholesterol homeostasis. Here are some comments:

  1. The link between snoRNA-jou and BAALC remains correlative. It's unclear whether BAALC is directly regulated by snoRNA-jou or affected indirectly via global RNA processing or chromatin effects. Clarify whether the snoRNA binds or modulates BAALC mRNA directly or indirectly.

The reviewer raises a very important and interesting point concerning the precise molecular function of the snoRNA. However, like the majority of the published articles of the literature, the direct link between the snoRNA and the associated deregulated genes is rarely directly demonstrated. This direct demonstration relies on tricky experiments as Ribosome or Polysome Profiling, whose are long range experiments and beyond the scope of this article.    

2.Figures 4 and 5a both compare sh_jou with shControl, but the total number of genes reported differs between them. Could you please clarify the reason for this discrepancy? It is currently unclear and somewhat confusing.

The Figures 4 and 5a are two different displays of the results of the RNA-Seq. The Figure 4 show that 726 genes are only expressed in sh-jou while 433 genes are only expressed in control (sh-control), while finally 10904 genes are commun to the 2 groups. Thus, this Venn diagramm does not give any information about if these genes are differentially expressed or not. Then, this is why these numbers are different than the number of DEG (Differentially Expressed Genes) presented in Figure 5a. To be brief, the Figure 4 is not directly associated and linked with the DEG.

Reviewer 3 Report

Comments and Suggestions for Authors

In this manuscript titled “Knockdown of the snoRNA-Jouvence Blocks the 2 Cell Proliferation and Leads to Cell Death of Human Primary 3 Glioblastoma Cells”, Jaque-Cabrera et. al., explore the effects of knock-down of snoRNA-jouvence (snoRNA-jou) in various cell lines, including primary Glioblastoma (GBM) cells. Role of snoRNAs in different types of cancers is an emerging field of interest and in this article the authors chose to study the role of a snoRNA previously studied by them in other systems and cell types. They used lentivirus-based vectors to express short-hairpin RNAs (shRNAs) targeting the snoRNA-jou in different cell lines including PDX-derived GBM cells. They have employed various methods to study the effects on cell proliferation and viability, metabolism-based as well as DNA-based. In both, they observed decreased proliferation after knocking down the snoRNA-jou. In addition, they also observed increase in apoptotic cells. The authors went own to perform an unbiased approach to investigate changes in transcriptome after knockdown of the snoRNA-jou in GBM14 cells. Eventually they identified various target genes and pathways including BAALC and links to cholesterol metabolism.

Overall, the authors have presented a well-structured analysis and the manuscript is fairly well written. I have a few minor concerns and suggestions which are stated below:

1)The authors state the lentiviral vector used is “newly designed” but it is not adequately described in the method section. It would be good to describe the vector information.

2) The supplementary figure 1 was not provided. But as I understand it efficacy of the shRNA targeting snoRNA-jou in HCT116 was previously performed. Also did the authors try more than one shRNA before choosing one? If so, this could be included.

3) The authors have very nicely presented the Figure 1 panel, especially including the percent changes, number of samples/times etc. is informative and easy to read. But the panels could be rearranged for e.g., all HCT116 data together followed by other cell types and/or different methods used together etc. Grouping and presenting the knock-down (KD) efficiency in different cell types as initial panel might be better.

4) It would good if the authors could include bright field images alongside the “microphotography” panels. In addition, the eGFP images of the cells as stated could be beneficial.

5) The EdU/TUNEL studies presented are at day 3 while the maximum KD was between 8-10 days. Why have the authors chosen to perform the EdU/TUNEL at this earlier time point?

6) Overall, the article is exploratory and suggests several pathways of interest based on correlation mainly through previous findings. The authors also use Gene Ontology analysis to derive conclusions here. They should describe the GO analysis in detail in the methods section. For example, the gene list used, and criterion for the shortlist if any etc. Especially since the terms enriched do not immediately correspond to the pathways or functions discussed in detail later, including the cholesterol metabolism. The genes involved in these pathways are not the ones that showed the highest regulation (fold change and significance).

7) Methods part could be more descriptive, including transcriptomic analysis, GO analysis, staining for counting experiments etc.

8) There are several typos and phrases that need modification. The text would very much benefit from thorough proof reading.

Comments on the Quality of English Language

Few of the instances showing need for improvement in text/language :

Line 31- "potentially be use as a"

Line 129- "Then, we decide to perform"

Line 165- "above, we investigate the sh-lentivirus "

Line 168- "we observe a "

Line 180-181-"whose are also not integrated in the genome".. etc.

Author Response

In this manuscript titled “Knockdown of the snoRNA-Jouvence Blocks the 2 Cell Proliferation and Leads to Cell Death of Human Primary 3 Glioblastoma Cells”, Jaque-Cabrera et. al., explore the effects of knock-down of snoRNA-jouvence (snoRNA-jou) in various cell lines, including primary Glioblastoma (GBM) cells. Role of snoRNAs in different types of cancers is an emerging field of interest and in this article the authors chose to study the role of a snoRNA previously studied by them in other systems and cell types. They used lentivirus-based vectors to express short-hairpin RNAs (shRNAs) targeting the snoRNA-jou in different cell lines including PDX-derived GBM cells. They have employed various methods to study the effects on cell proliferation and viability, metabolism-based as well as DNA-based. In both, they observed decreased proliferation after knocking down the snoRNA-jou. In addition, they also observed increase in apoptotic cells. The authors went own to perform an unbiased approach to investigate changes in transcriptome after knockdown of the snoRNA-jou in GBM14 cells. Eventually they identified various target genes and pathways including BAALC and links to cholesterol metabolism.

Overall, the authors have presented a well-structured analysis and the manuscript is fairly well written. I have a few minor concerns and suggestions which are stated below:

1) The authors state the lentiviral vector used is “newly designed” but it is not adequately described in the method section. It would be good to describe the vector information.

We have added a more detailed description of the vector in M&M, as well as we have added the Vector Map in Suppl. Figure 1a.

2) The supplementary figure 1 was not provided. But as I understand it efficacy of the shRNA targeting snoRNA-jou in HCT116 was previously performed. Also did the authors try more than one shRNA before choosing one? If so, this could be included.

Yes, sorry, I missed to upload the 3 Suppl. Figures when I have submitted the manuscritp (I apologizes). In fine, I have added 2 Suppl. Figures, then the manuscript contains 5 Supplementary Figures.

Concerning to try and use another sh-RNA, we did not generate a second sh-RNA. The main  reason relies and refers to our previous article (El-Khoury et al., BMC-Genomic, 2020), in which we have generated and try two different siRNAs, but it turned-out that only one works properly. Then we have chosen the same sequence (the one that works) to perform our sh-RNA-lentivirus. I also want to remind that the snoRNA-jouvence is only 150bp length, thus it is difficult to find a 20 bp sequence that did not have any other partial homology (off-target). Then our choice of target sequence is quite limited.

3) The authors have very nicely presented the Figure 1 panel, especially including the percent changes, number of samples/times etc. is informative and easy to read. But the panels could be rearranged for e.g., all HCT116 data together followed by other cell types and/or different methods used together etc. Grouping and presenting the knock-down (KD) efficiency in different cell types as initial panel might be better.

O.K., I have modified the Figure 1 accordingly. Thanks, it is more clear like that ! 

4) It would good if the authors could include bright field images alongside the “microphotography” panels. In addition, the eGFP images of the cells as stated could be beneficial.

O.K., See the Suppl. Figure 2 for the eGFP images, showing the efficacy of the sh-lentivirus transduction (again sorry, in the previous submission, the Suppl. Figures were missed).

We have added few bright field images in a new Suppl. Figures 3.

5) The EdU/TUNEL studies presented are at day 3 while the maximum KD was between 8-10 days. Why have the authors chosen to perform the EdU/TUNEL at this earlier time point?

We have volontarily chosen to perform the EdU and TUNEL at day-3 post-transduction, because after 8 to 10 days, almost all the cells are disapppeared (it remains almost no cells) (10 to 15%). Therefore we estimate that was better to perform the EdU and TUNEL at an earlier time point, at a moment when we still have a lot of cells and several of them are in the dying process (for TUNEL) or still in proliferation (for EdU). We think that it gives a better representativity of the reality, otherwise at 8-10 days, the “game is over”.  Similar for the RNA-Seq.

6) Overall, the article is exploratory and suggests several pathways of interest based on correlation mainly through previous findings. The authors also use Gene Ontology analysis to derive conclusions here. They should describe the GO analysis in detail in the methods section. For example, the gene list used, and criterion for the shortlist if any etc. Especially since the terms enriched do not immediately correspond to the pathways or functions discussed in detail later, including the cholesterol metabolism. The genes involved in these pathways are not the ones that showed the highest regulation (fold change and significance).

O.K. I have added more information in M&M for the GO Ontology Analysis.

Moreover, it is true that the genes that I have put in front of the analysis (Tables 1 & 2) , as the one’s in cholesterol pathways are not necessarily the most deregulated one’s, but I have tried to put in front some genes that we could show an effect on the cells biology (here the lipid and cholesterol). I agree that several genes are deregulated, but as with any RNA-Seq outcome, we have to choose some of them and demonstrates that it has a biological meaning to that.  

7) Methods part could be more descriptive, including transcriptomic analysis, GO analysis, staining for counting experiments etc.

O.K., done (see the text) in M.&M.  

8) There are several typos and phrases that need modification. The text would very much benefit from thorough proof reading.

Comments on the Quality of English Language

Few of the instances showing need for improvement in text/language :

Line 31- "potentially be use as a"    “used”

Line 129- "Then, we decide to perform"   “we perform”

Line 165- "above, we investigate the sh-lentivirus "    “investigated”

Line 168- "we observe a "          “a decrease of proliferation of 52%”

Line 180-181-"whose are also not integrated in the genome".. etc.   “which are not”

Round 2

Reviewer 1 Report

Comments and Suggestions for Authors

Authors have improved the manuscript flow and tried to remedy as much as possible. I have no additional comments. 

Reviewer 2 Report

Comments and Suggestions for Authors

This version is fine to me.